# ML277 specifically enhances the fully activated open state of KCNQ1 by modulating VSD-pore coupling

Panpan Hou[1,2,3], Jingyi Shi[1,2,3], Kelli McFarland White[1,2,3], Yuan Gao[4], Jianmin Cui[1,2,3]*

[1]Department of Biomedical Engineering, Washington University, St. Louis, United States; [2]Center for the Investigation of Membrane Excitability Disorders, Washington University, St. Louis, United States; [3]Cardiac Bioelectricity and Arrhythmia Center, Washington University, St. Louis, United States; [4]Tencent AI Lab, Shenzhen, China

**Abstract** Upon membrane depolarization, the KCNQ1 potassium channel opens at the intermediate (IO) and activated (AO) states of the stepwise voltage-sensing domain (VSD) activation. In the heart, KCNQ1 associates with KCNE1 subunits to form $I_{Ks}$ channels that regulate heart rhythm. KCNE1 suppresses the IO state so that the $I_{Ks}$ channel opens only to the AO state. Here, we tested modulations of human KCNQ1 channels by an activator ML277 in *Xenopus* oocytes. It exclusively changes the pore opening properties of the AO state without altering the IO state, but does not affect VSD activation. These observations support a distinctive mechanism responsible for the VSD-pore coupling at the AO state that is sensitive to ML277 modulation. ML277 provides insights and a tool to investigate the gating mechanism of KCNQ1 channels, and our study reveals a new strategy for treating long QT syndrome by specifically enhancing the AO state of native $I_{Ks}$ currents.

DOI: https://doi.org/10.7554/eLife.48576.001

*For correspondence:
jcui@wustl.edu

## Introduction

KCNQ1, also known as $K_V7.1$ or $K_VLQT1$, belongs to a subfamily of voltage-activated potassium channels (*Barhanin et al., 1996*; *Sanguinetti et al., 1996*; *Wang et al., 1996*). It adopts the canonical structural organization of the homo-tetrameric $K_V$ superfamily, in which each subunit contains six transmembrane segments (S1-S6), with the first four (S1-S4) forming the voltage-sensing domain (VSD) and the latter two (S5-S6) folding to the pore (*Sun and MacKinnon, 2017*). In response to membrane depolarization, the S4 segment in the VSD undergoes outward movement (*Glauner et al., 1999*; *Larsson et al., 1996*) and triggers pore opening via electromechanical (E-M) coupling interactions between the VSD and the pore. Similar to other $K_V$ members such as the Shaker channel, the VSD of KCNQ1 channels shows two steps activation: it activates sequentially from the resting state to an experimentally resolvable intermediate state before finally arriving at the fully activated state (*Baker et al., 1998*; *Barro-Soria et al., 2014*; *Bezanilla et al., 1994*; *Carvalho-de-Souza and Bezanilla, 2018*; *Delemotte et al., 2011*; *Hou et al., 2017*; *Jensen et al., 2012*; *Lacroix et al., 2012*; *Ledwell and Aldrich, 1999*; *Osteen et al., 2012*; *Sigg et al., 1994*; *Sigworth, 1994*; *Silva et al., 2009*; *Silverman et al., 2003*; *Wu et al., 2010a*; *Wu et al., 2010b*; *Zagotta et al., 1994*; *Zaydman et al., 2014*). However, unlike the Shaker channel which opens only when the VSD moves to the fully activated state, the KCNQ1 channel opens when its VSD occupies both the intermediate and fully activated conformations, resulting in intermediate-open (IO) and activated-open (AO) states (*Hou et al., 2017*; *Zaydman et al., 2014*).

The two open states, IO and AO, are distinctive from each other. Structurally, they associate with different VSD and pore conformations. During membrane depolarization, the S4 segment moves outward from its resting state to the intermediate state to trigger the IO state opening. Subsequent movement of the S4 helix from the intermediate state to the activated state then triggers conformational change of the pore from the IO state to the AO state. These VSD-pore conformations can be isolated by the mutation that arrests the VSD at the intermediate state, E160R/R231E (E1R/R2E), which results in the constitutive IO state, and the mutation that arrests the VSD at the activated state, E160R/R237E (E1R/R4E), which results in the constitutive AO state (*Restier et al., 2008*; *Wu et al., 2010a*; *Wu et al., 2010b*; *Zaydman et al., 2014*). Functionally, the two open states have different gating properties and are differentially modulated by the auxiliary subunit KCNE1, allowing us to experimentally distinguish them. These differences include (1) the AO state shows slower current kinetics and more positive voltage-dependent activation than that of the IO state, (2) KCNQ1 channels show lower open probability when the VSD moves from the intermediate state to the fully activated state, resulting in inactivation phenotypes under certain voltage stimulations, (3) the AO state shows lower $Rb^+/K^+$ permeability ratio than the IO state, (4) different E-M coupling mechanisms such that the two open states can be isolated by single mutations that ablate respective E-M couplings, and (5) KCNE1 suppresses the IO state but enhances the AO state (*Hou et al., 2017*; *Zaydman et al., 2014*).

In cardiomyocytes, KCNQ1 associates with KCNE1 to generate the slow delayed rectifier current ($I_{Ks}$) important for terminating cardiac action potentials (*Barhanin et al., 1996*; *Chiamvimonvat et al., 2017*; *Keating and Sanguinetti, 2001*; *Sanguinetti et al., 1996*). Clinically, more than 300 loss-of-function mutations in KCNQ1 have been found to induce functional defects to the $I_{Ks}$ current, causing long QT syndrome (LQTS) that predisposes patients to life-threatening cardiac arrhythmia (*Schwartz et al., 2012*; *Wang et al., 1996*). KCNE1 specifically suppresses the IO state but enhances the AO state, rendering the $I_{Ks}$ channel with only AO state openings (*Zaydman et al., 2014*). Therefore, the AO state is physiologically more important in the heart. Enhancing AO state currents to specifically rescue the functional defects of the mutant $I_{Ks}$ currents presents a potential strategy for anti-LQTS therapy.

Different activators have been found to enhance KCNQ1 and $I_{Ks}$ currents. Zinc pyrithione (ZnPy), L-364,373 (R-L3), 4,4′-diisothiocyanatostilbene-2,2′-disulfonic acid (DIDS), and mefenamic acid (MFA) activate either homomeric KCNQ1 or KCNQ1 +KCNE1 complex (*Abitbol et al., 1999*; *Busch et al., 1997*; *Gao et al., 2008*; *Salata et al., 1998*; *Seebohm et al., 2003a*; *Xiong et al., 2007*), while phenylboronic acid (PBA), Hexachlorophene (HCP), mallotoxin (MTX), and 3-ethyl-2-hydroxy-2-cyclopenten-1-one (CPT1) potentiate both KCNQ1 and KCNQ1 +KCNE1 channels (*De Silva et al., 2018*; *Mruk and Kobertz, 2009*; *Zheng et al., 2012*). Although some of these activators are capable of shortening the action potential duration (APD) in cardiac myocytes (*Salata et al., 1998*; *Zheng et al., 2012*), none of them was found to only enhance the AO state of KCNQ1 or $I_{Ks}$ channels. Besides KCNQ1 and $I_{Ks}$ channels, all these activators also strongly modulate other ion channels that are key to cardiac electrophysiology, such as hERG, $Ca_V1.2$, and $Na_V1.5$ in the heart and neuronal KCNQ channels (KCNQ2-KCNQ5) (*Cruickshank et al., 2003*; *De Silva et al., 2018*; *Gao et al., 2017*; *Hill and Sitsapesan, 2002*; *Mruk and Kobertz, 2009*; *Salata et al., 1998*; *Zheng et al., 2012*). The low specificity may cause severe side effects to LQTS patients, limiting the potential for further development into anti-LQTS drugs.

On the other hand, a recently identified KCNQ1 activator ML277 shows potent effect to KCNQ1 channel, while demonstrating little effects on other important cardiac ion channels including hERG, Nav1.5, and Cav1.2 channels, and other KCNQ isoforms (*Yu et al., 2013*; *Yu et al., 2010*). Moreover, ML277 shortens the APD of human-induced pluripotent stem cell (iPSC)-derived cardiomyocytes and guinea pig cardiomyocytes, via activating the $I_{Ks}$ currents (*Xu et al., 2015*; *Yu et al., 2013*). The high specificity and potent potentiation in native $I_{Ks}$ currents make ML277 a promising anti-LQTS drug candidate. Although computational and experimental studies have been performed to probe the binding and potentiation of ML277 on KCNQ1 and $I_{Ks}$ channels (*Xu et al., 2015*; *Yu et al., 2013*), the mechanism underlying this potentiation on the two open states of KCNQ1 channels remains largely unclear.

In this study, we find that ML277 changes KCNQ1 channel current kinetics, alters voltage dependence of activation, inactivation property, and ion permeation. All these modifications are consistent with the mechanism that ML277 specifically enhances the current of the AO state. Supporting this

mechanism, ML277 selectively enhances the currents of mutant KCNQ1 channels that open only at the AO state but not the mutant channels that open only at the IO state. These results and recordings of VSD activation using voltage clamp fluorometry (VCF) suggest that ML277 specifically promotes the E-M coupling when the VSD is at the activated state to increase the AO state current. KCNQ1 is the first ion channel that shows well resolved two open states when the VSD is at different activation states. This study sets the first example that a small molecule compound can specifically modulate only one of the two open states via enhancing the specific E-M coupling. Our study not only proposes an effective tool to investigate the gating mechanism of KCNQ1 channel with two open states, but also paves the way to new strategies, enhancing the AO state of native $I_{Ks}$ currents, for treating LQTS.

## Results

### ML277 specifically enhances currents of the AO state

Previous studies have shown that, in Chinese hamster ovary (CHO) cells, ML277 potentiates both KCNQ1 channels and KCNQ1 +KCNE1 channels with unsaturated KCNE1 association. Progressive increase of KCNE1 expression reduces efficacy of ML277 and eventually abolishes its effect on KCNQ1 +KCNE1 channels with saturated KCNE1 binding (*Yu et al., 2013*). We tested the ML277 effects on KCNQ1 and $I_{Ks}$ channels expressed in *Xenopus* oocytes, and found that 1 µM ML277 increased the KCNQ1 current amplitudes, such that the same depolarizing voltage elicited larger currents (*Figure 1A*), while with 1 µM ML277 less current increase was shown in KCNQ1 +KCNE1 channels with injected mRNA weight ratio of KCNQ1:KCNE1 = 300:1, which may induce unsaturated KCNE1 binding, and no current increase was shown in KCNQ1 +KCNE1 channels with injected

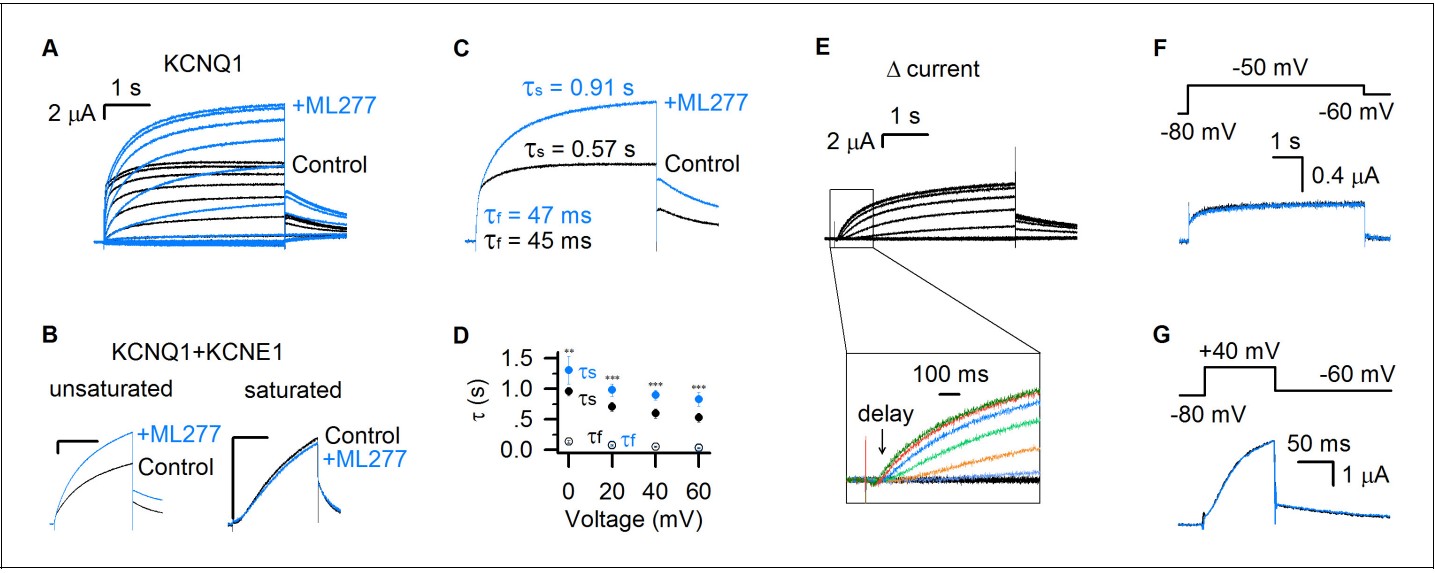

**Figure 1.** ML277 specifically changes the slow component of KCNQ1 currents. (**A**) KCNQ1 currents from *Xenopus* oocytes recorded before (black) and after (blue) adding 1 µM ML277. The test voltages were from −120 mV to 80 mV with 20 mV increments, and then returned to −40 mV for measuring the tail currents. (**B**) 1 µM ML277 effects on KCNQ1 +KCNE1 channels with injected mRNA weight ratio of KCNQ1:KCNE1 = 300:1 (left, unsaturated KCNE1 binding) and 4:1 (right, saturated KCNE1 binding). The test voltages were 40 mV for 4 s, and then returned to −40 mV. (**C**) Activation time constants (τ) of KCNQ1 currents recorded at +40 mV before (black) and after (blue) adding ML277. The time constants of the fast component ($τ_f$) and slow component ($τ_s$) were 48 ± 3 ms and 0.60 ± 0.03 s for control, and 52 ± 3 ms and 0.99 ± 0.07 s for after adding ML277. The p values were 0.37 for $τ_f$, and 0.00091 for $τ_s$, t-test, n ≥ 4. (**D**) Averaged results of activation time constants ($τ_f$ and $τ_s$) of KCNQ1 currents recorded at different voltages before (black) and after (blue) adding ML277. n ≥ 4. (**E**) The increased current by ML277 ($I_{ML277}$-$I_{Control}$, Δ current) from panel A. The first 1 s activation currents were shown with insets of enlarged scales, and different colors were utilized to show the 'delay' of the increased currents at different voltages. (**F–G**) KCNQ1 currents recorded before (black) and after (blue) adding ML277 with 5 s time duration at −50 mV (**F**) and 100 ms time duration at +40 mV (**G**). The tail currents were recorded at −60 mV. Voltage protocols are shown on the top. All error bars in this and other figures are ± SEM.
DOI: https://doi.org/10.7554/eLife.48576.002

mRNA weight ratio of 4:1, which may saturate the KCNE1 binding (*Figure 1B*) (*Nakajo et al., 2010*). These results recorded from *Xenopus* oocytes are consistent with previous findings from CHO cells (*Yu et al., 2013*). The loss of ML277 effects on KCNQ1 +KCNE1 channels with saturating KCNE1 association was suggested to result from a competition between ML277 and KCNE1 in binding to KCNQ1 (*Yu et al., 2013*). ML277 may bind at the interface between the VSD and the pore from two neighboring subunits, where KCNE1 is also found to bind with KCNQ1 (*Chan et al., 2012*; *Nakajo et al., 2010*; *Peng et al., 2017*; *Ramasubramanian and Rudy, 2018*; *Strutz-Seebohm et al., 2011*; *Xu and Rudy, 2018*). As a result, KCNE1 binding may block ML277 binding and abrogate its effects.

The KCNQ1 currents can be fitted with two distinct exponential components in response to depolarization pulses (*Figure 1A*) (*Hou et al., 2017*). The fast time course approximates the channels opening to the IO state, while the slow component is mainly determined by the channels opening to the AO state (*Hou et al., 2017*). ML277 increased only the slow current component, while leaving the fast current component unaltered, which is clearly shown by merging the currents before and after ML277 application (*Figure 1A,C*). We compared the time constants of the fast and slow components before and after ML277 application. For example, at +40 mV depolarization, the time constant of the slow component ($\tau_s$) was changed from 0.57 s to 0.91 s, while the time constant of the fast component ($\tau_f$) changed little (from 45 ms to 47 ms, *Figure 1C*). This selective change on the $\tau_s$ was observed in different voltages (*Figure 1D*). We also analyzed the increased currents ($\Delta$ currents) by subtracting control currents from currents in the presence of ML277 (*Figure 1E*). It is clear that the $\Delta$ currents have no fast current component any more and show different delays at different voltages, following the onset of the IO state (*Figure 1A,E*). A change only in time constant and amplitude of the slow component is quantitatively consistent with the mechanism that ML277 selectively alters the AO state without affecting the IO state.

Our previous studies showed that the IO and AO states have different time- and voltage-dependence: the AO state appears only when the KCNQ1 channel is activated by voltages above −40 mV and for a sufficient time duration ($\geq$100 ms at +40 mV) (*Hou et al., 2017*). To verify that ML277 selectively increases the currents of the AO state, we tested the effect of 1 μM ML277 on KCNQ1 current when we activated KCNQ1 channels either at low voltage (−50 mV) or at high voltage with a short time duration (+40 mV for only 100 ms). These voltage protocol effectively restricts KCNQ1 channel activation to the IO state. In both cases, ML277 did not increase current amplitudes or alter current kinetics (*Figure 1F,G*). These data support that ML277 specifically modifies the current of the AO state.

## ML277 right shifts the G–V relationship of KCNQ1 channels

We analyzed ML277-induced changes in the voltage-dependent activation of KCNQ1 channels. ML277 activated the channel and nearly doubled the current size and the total conductance (*Figures 1A* and *2A*). Interestingly, unlike other activators that potentiate ion channel currents by left-shifting the G–V relationship (*De Silva et al., 2018*; *Gao et al., 2017*; *Guo et al., 2017*; *Liu et al., 2013*; *Seebohm et al., 2003a*; *Zheng et al., 2012*), ML277 slightly shifted the G–V relationship of KCNQ1 channels to more positive voltages ($\Delta$V ~ 6 mV, *Figure 2B,C*). This curious result can be explained by properties of the KCNQ1 voltage-dependent activation process. The KCNQ1 currents are composed of currents of both the IO and AO states, therefore, the G–V relation of KCNQ1 channels is the sum of the G–V relations of both the IO and AO states. The G-V relationship of the AO state is at more positive voltages compared to the IO state, reflecting the fact that the activated VSD state requires more positive voltages than the intermediate VSD state (*Hou et al., 2017*; *Zaydman et al., 2014*). Our previous studies demonstrated that KCNQ1 conducts current predominately through the IO state (*Hou et al., 2017*; *Zaydman et al., 2014*). This property is also manifested in the kinetics of current activation, in which the slow component (primarily through the AO state) accounts for only a small portion of the total current as compared to the fast component (primarily through the IO state) (*Figure 1A*). The G–V relation of KCNQ1 channels thus tracks closely to the G–V relation of the IO state, and contains a small portion of the G–V relation of the AO state. By contrast, the $I_{Ks}$ (KCNQ1 +KCNE1) channel conducts predominantly with AO-state current, because KCNE1 suppresses pore conductance of the IO state (*Zaydman et al., 2014*). The G-V curve of the $I_{Ks}$ channel is thus dominated by the G–V relationship of the AO state, and is significantly right-shifted compared to the KCNQ1 G-V relationship (*Figure 2C,D*). The contrasting G-V relations

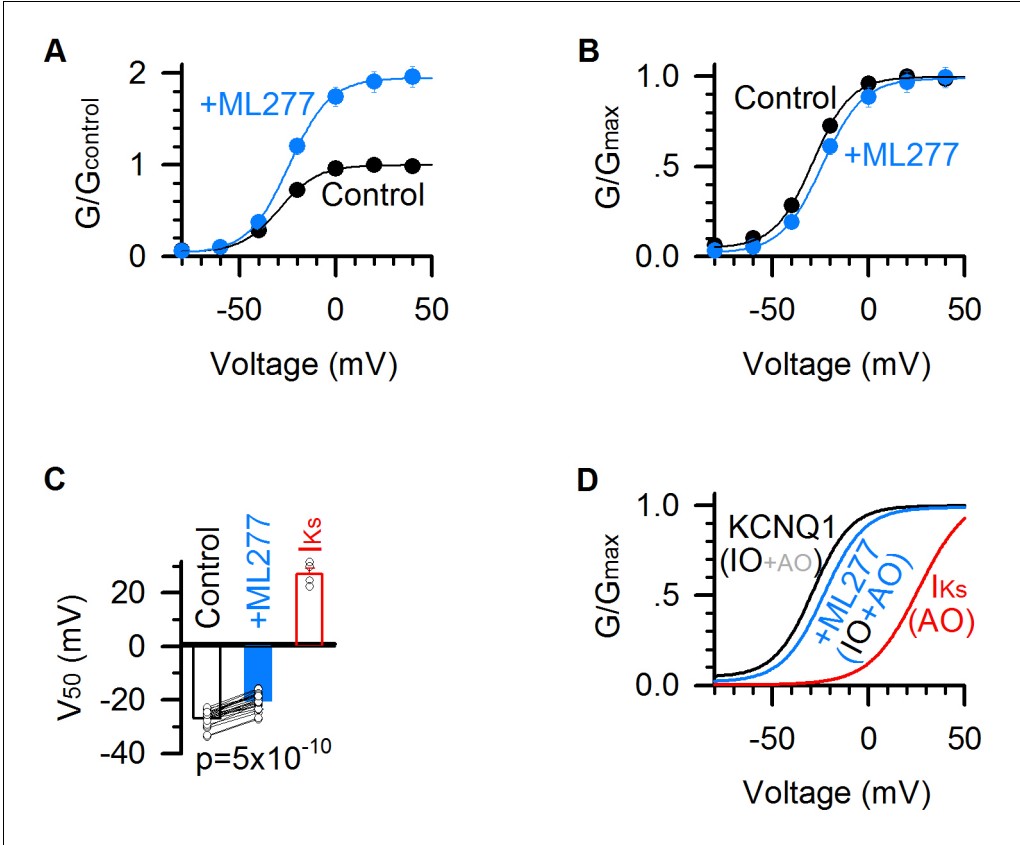

**Figure 2.** ML277 right shifts the G–V relationship of KCNQ1 channels. (**A**) G–V relationships of KCNQ1 channels before (black) and after (blue) adding 1 μM ML277. Data points were fitted with a Boltzmann function. n = 25. (**B**) Normalized G–V relationships from panel A. (**C**) Averaged results of $V_{50}$ of G–V relations (voltage at which the G–V is half maximal). The $V_{50}$ for KCNQ1 channels is $-26.8 \pm 0.6$ mV in control (black), and $-20.3 \pm 0.6$ mV after adding ML277 (blue), with a significant difference (p=$5 \times 10^{-10}$, paired t test. n = 25). The $V_{50}$ of $I_{Ks}$ channels (KCNQ1 +KCNE1) ($25.6 \pm 3.1$ mV, n = 4, red) is also for comparison. (**D**) G–V relationships of the KCNQ1 channels before (black) and after (blue) adding 1 μM ML277, and of $I_{Ks}$ channels (red). IO/AO composition in each type of channels is indicated in parentheses.

DOI: https://doi.org/10.7554/eLife.48576.003

of the KCNQ1 (IO-dominant) and $I_{Ks}$ (AO-dominant) are illustrated in *Figure 2C and D*. In this study, we utilized a single Boltzmann function, which assumes one closed state and one open state, to approximate the complex KCNQ1 G–V relation (*Figure 2A,B*). Accordingly, if ML277 increases the contribution of the AO state to the overall current, then a right-shift in the G-V relationship is expected, as the AO state is right-shifted compared to the IO state. We indeed observed a small right-shift in the KCNQ1 G-V relationship upon ML277 application (*Figure 2B,C*). The right shift of the G–V relation by ML277 is consistent with the mechanism that ML277 selectively increases the AO state currents and thereby increases contribution of the AO state G-V relation to the overall G-V curve.

## ML277 changes the inactivation of KCNQ1 channels

KCNQ1 channels show an incomplete inactivation that can be observed by a hook in tail currents at hyperpolarized voltages after a pre-pulse that activates the channel (*Abitbol et al., 1999*; *Gibor et al., 2007*; *Larsen et al., 2011*; *Tristani-Firouzi and Sanguinetti, 1998*). The hook is due to an initial increase of the inward (downward) tail current followed by continuous decay of the current (*Figure 3A*), consistent with the idea that channels recover from inactivation upon hyperpolarization via an open state before closing due to deactivation (*Gibor et al., 2007*; *Larsen et al., 2011*; *Pusch et al., 1998*; *Seebohm et al., 2003b*). Recently, we found that the so-called inactivation state

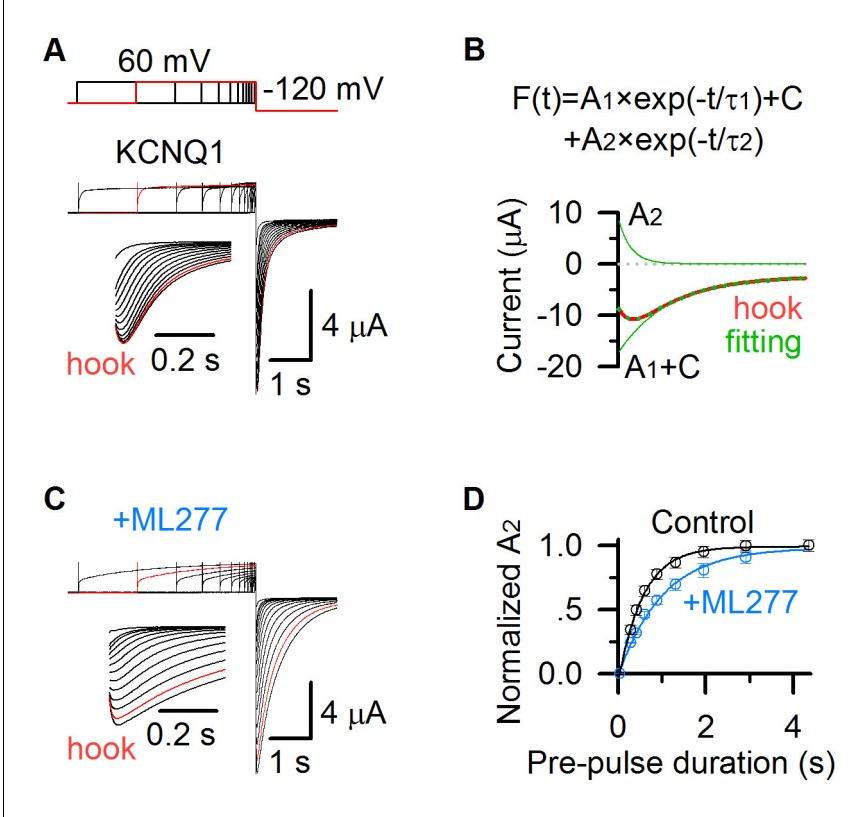

**Figure 3.** ML277 changes inactivation of KCNQ1 channels. (**A**) KCNQ1 tail currents recorded with pre-pulses of increasing time duration in 100 mM K$^+$ solution. The pre-pulses were +60 mV with time durations ranging between 0.03 and 4.355 s, and the test pulse was −120 mV for 2 s. The inset shows the hook in tail currents with an expanded time scale. The current recorded with a 2.913 s pre-pulse duration is shown in red. (**B**) KCNQ1 tail current recorded with a 2.913 s pre-pulse duration (red) was fitted with a double exponential function $F(t) = A_1 \times \exp(-t/\tau_1) + A_2 \times \exp(-t/\tau_2) + C$ (**Hou et al., 2017**). The fitting curve is shown in dotted green line, the deactivation component ($A_1 \times \exp(-t/\tau_1) + C$) and hook component ($A_2 \times \exp(-t/\tau_2)$) are shown in green lines. (**C**) KCNQ1 tail currents after applying ML277 recorded with pre-pulses of increasing time durations in 100 mM K$^+$ solution, with the same voltage protocol shown as in panel A. The inset shows the hook in tail currents with an expanded time scale. The current recorded with a 2.913 s pre-pulse duration is shown in red. (**D**) Normalized $A_2$ of KCNQ1 hook currents before (black) and after (blue) adding ML277 vs. time durations of the pre-pulse ($n \geq 4$). Data are fitted with a single exponential function.

DOI: https://doi.org/10.7554/eLife.48576.004

in KCNQ1 channels is the AO state, and the hook current reflects channel deactivation from the AO state to the IO state because the channels have a higher open probability when the VSD is at the intermediate state than at the activated state (**Hou et al., 2017**). If ML277 selectively enhances the currents through the AO state, we expect to observe a change in KCNQ1 hook currents.

We recorded tail currents during hyperpolarization to −120 mV, with varying time durations of pre-pulses at +40 mV that activated KCNQ1 channels. The hook current developed with pre-pulse time durations, and saturated at the 2.9 s pre-pulse (red, **Figure 3A**). To quantify the inactivation, we fitted the tail currents with a double exponential function $F(t) = A_1 \times \exp(-t/\tau_1) + A_2 \times \exp(-t/\tau_2) + C$ (**Figure 3B**), where $A_1$ and $A_2$ are amplitudes, $\tau_1$ and $\tau_2$ are time constants, and $C$ is the offset due to leak currents. The hook current, recorded after a pre-pulse of 2.913 s duration at +40 mV (red, **Figure 3A,B**), was well fitted by the double exponential function (green dots, **Figure 3B**). $A_1 \times \exp(-t/\tau_1) + C$ represents the deactivation process (green line, **Figure 3B**), and $A_2 \times \exp(-t/\tau_2)$ represents the hook current (green line, **Figure 3B**). $A_2$ is the amplitude of the hook current, which is used to quantify the KCNQ1 inactivation. The normalized $A_2$ before (black) and after (blue) adding ML277 at varying pre-pulse time durations is shown in **Figure 3D**. It shows that ML277 slows down

the development of $A_2$ (from 0.56 s to 1.02 s, *Figure 3D*), consistent with the effect that ML277 prolongs the development of the AO state (from 0.57 s to 0.91 s, *Figure 1C*). These results reveal that ML277 modulates the AO state and thus the inactivation property.

## ML277 changes ion permeation of KCNQ1 channels

Previous studies found that KCNQ1 and $I_{Ks}$ channels have different ion permeation. The inward tail currents of KCNQ1 channels at hyperpolarized voltages are increased about threefold upon substitution of 100 mM potassium with 100 mM rubidium (*Pusch et al., 2000*), however, KCNE1 association reverses this change and makes the $I_{Ks}$ channel conducts less tail current in 100 mM rubidium than in 100 mM potassium (*Pusch et al., 2000*; *Seebohm et al., 2003b*). We recently found that these results could be explained by the mechanism that the IO and AO states have different $Rb^+/K^+$ permeability ratio (*Zaydman et al., 2014*). Using E1R/R2E and E1R/R4E mutations that open in the IO and AO state, respectively, we found that E1R/R4E had a significantly lower $Rb^+/K^+$ permeability ratio compared to E1R/R2E channels, indicating that the AO state has a lower $Rb^+/K^+$ permeability ratio than the IO state (*Zaydman et al., 2014*). Since KCNQ1 channels open predominately to the IO state, they show a high $Rb^+/K^+$ permeability ratio, similar to that of the IO state. On the other hand, since KCNE1 suppresses the IO state but enhances the AO state, the $I_{Ks}$ channels show a lower $Rb^+/K^+$ permeability ratio (*Zaydman et al., 2014*).

To test ML277 induced changes in ion permeation of KCNQ1 channels, we measured tail current amplitudes of the channels before and after adding 1 μM ML277 at −60 mV after a + 60 mV pre-pulse for 5 s in 100 mM $Rb^+$ and 100 mM $K^+$ solutions, respectively. ML277 significantly decreased the $Rb^+/K^+$ permeability ratio of KCNQ1 channels (from 3.1 to 1.1, *Figure 4*). The $Rb^+/K^+$ permeability ratio of KCNQ1 channels after treating ML277 is closer to that of the $I_{Ks}$ channels (*Figure 4B*), which is consistent with previous finding (*Xu et al., 2015*). These results are consistent with the mechanism that ML277 selectively enhances currents of the AO state to change the IO/AO composition in KCNQ1 channels, and the increased fraction of AO currents changes the $Rb^+/K^+$ permeability ratio toward that of the AO state.

## ML277 specifically increases the currents of mutant KCNQ1 channels that open only to the AO state

In previous studies, we have demonstrated that the IO and AO states can be isolated in mutant KCNQ1 channels. Two pairs of mutations make the channels open exclusively to the IO and AO states with different mechanisms, respectively. The first pair, E1R/R2E and E1R/R4E arrest the VSDs at the intermediate and activated states, respectively, and make the channels constitutively open at the IO and AO states, respectively (*Restier et al., 2008*; *Wu et al., 2010a*; *Wu et al., 2010b*; *Zaydman et al., 2014*). The second pair, S338F and F351A disrupt the VSD-pore couplings when the VSDs are at the activated and intermediate states, respectively, and make the channels open

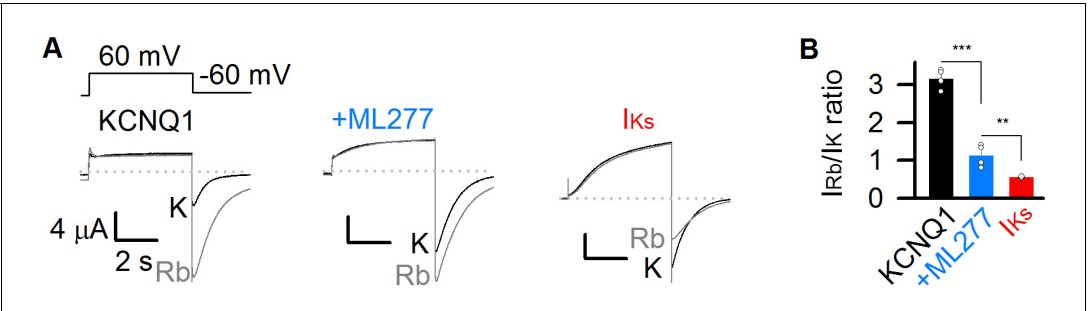

**Figure 4.** ML277 changes ion permeation of KCNQ1 channels. (**A**) Currents from KCNQ1 channels before (left) and after (middle) adding 1 μM ML277, as well as $I_{Ks}$ channels (KCNQ1 +KCNE1) (right) recorded in 100 mM $K^+$ (black) or 100 mM $Rb^+$ (gray) external solutions. The tail currents were elicited at −60 mV for 3 s from a pre-pulse of +60 mV for 5 s (top of left). (**B**) Averaged $Rb^+/K^+$ permeability ratios of tail current amplitudes for KCNQ1 before (3.1 ± 0.2, black) and after (1.1 ± 0.3) adding ML277, and for $I_{Ks}$ channels (0.6 ± 0.1, red). All n ≥ 4.
DOI: https://doi.org/10.7554/eLife.48576.005

only to the IO and AO states, respectively (*Boulet et al., 2007*; *Hoosien et al., 2013*; *Hou et al., 2017*; *Zaydman et al., 2014*).

To confirm that ML277 specifically enhances the AO state current, we measured its effects on these mutant channels. ML277 increased currents of the mutant channels E1R/R4E and F351A that open only in the AO state, but barely enhanced the mutant channels E1R/R2E or S338F that open only to the IO state (*Figure 5*). These results support the mechanism that ML277 selectively enhances the current of the AO state.

## ML277 enhances the E-M coupling specifically when the VSD is at the activated state

The above results all support the mechanism that ML277 selectively enhances the currents of the AO state in KCNQ1 channels. Since ML277 slows down the time course of both activation and deactivation (*Figures 1C* and *3A,C*), it is likely to modulate the gating mechanism of KCNQ1 that specifically controls the AO state. The voltage-dependent activation of a $K_V$ channel involves three molecular events: VSD activation upon transmembrane voltage depolarization, rearrangement in interactions between the VSD and the pore (E-M coupling), and the pore opening to conduct ionic current (*Cui, 2016*). Interestingly, ML277 enhances the constitutive current of E1R/R4E mutant channels (*Figure 5A,C*), in which the VSD is arrested at the activated state by the double charge reversal mutations (*Zaydman et al., 2014*). This result indicates that ML277 may enhance the coupling between the VSD and pore when the VSD is at the activated state to increase open probability.

To further study this mechanism, we carried out VCF experiments, a technique that VSD movements are monitored with a fluorophore attached to the S3-S4 linker in pseudo-WT KCNQ1 (KCNQ1-C214A/G219C/C331A) channels during activation, while channel opening is monitored by current recordings (*Barro-Soria et al., 2015*; *Barro-Soria et al., 2014*; *Hou et al., 2017*; *Osteen et al., 2012*; *Osteen et al., 2010*; *Zaydman et al., 2014*; *Zaydman et al., 2013*). The KCNQ1 fluorescence-voltage (F–V) relationship can be well-fitted by a double Boltzmann function, revealing the stepwise VSD activation processes (*Figure 6A,B*) (*Hou et al., 2017*; *Zaydman et al., 2014*). The time- and voltage-dependence of the VSD activation in the absence and presence of 1 µM ML277 showed little change, and the fluorescence traces could be superimposed with each other at different voltages (*Figure 6A,B*). Similar to the WT KCNQ1 channels (*Figure 2B*), ML277 induced a right-shifted G–V relationship in pseudo-WT KCNQ1 (ΔV ~ 5 mV, *Figure 6B*). The VCF

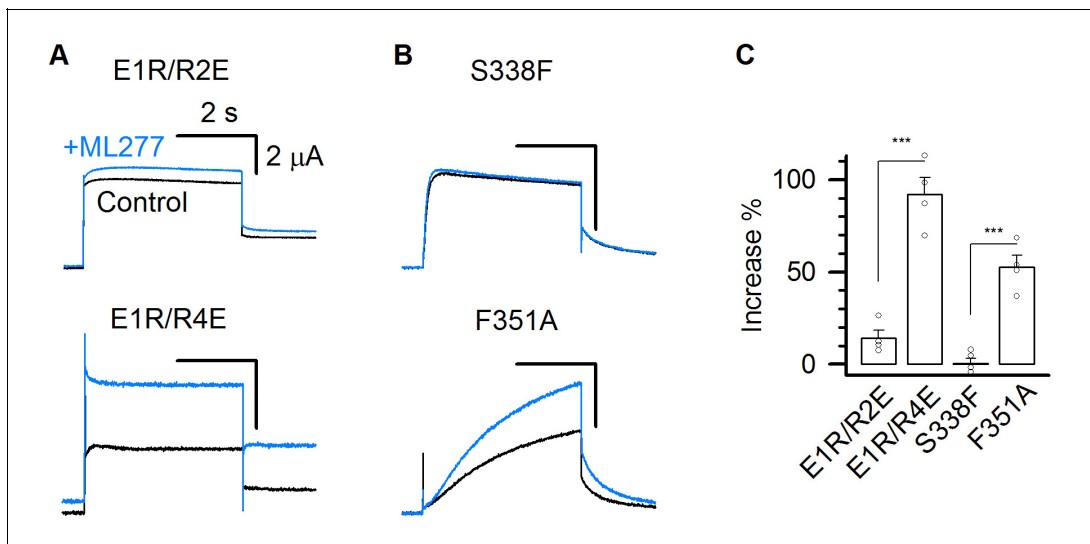

**Figure 5.** ML277 specifically increases the currents of mutant KCNQ1 channels that open only to the AO state. (A–B) 1 µM ML277 effects on E1R/R2E, E1R/R4E, S338F, and F351A mutant KCNQ1 channels. Currents were elicited at +40 mV for 4 s and then returned to −40 mV to record the tail currents. (C) Averaged percentage of ML277 induced current increase on E1R/R2E (14.2 ± 4.2%), E1R/R4E (95.5 ± 8.5%), S338F (0.14 ± 3.1%), and F351A (52.5 ± 6.3%). n ≥ 4.

DOI: https://doi.org/10.7554/eLife.48576.006

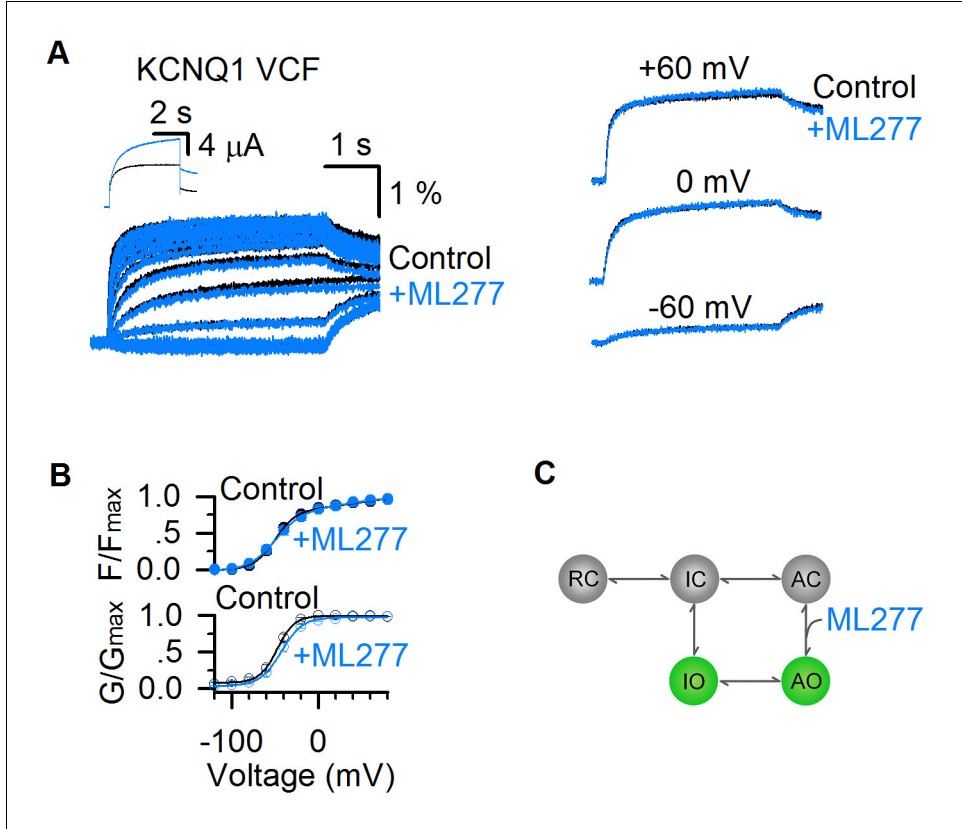

**Figure 6.** ML277 enhances currents of KCNQ1 channels but does not change VSD activation. (**A**) Fluorescence signals showing voltage sensor activation of pseudo-WT KCNQ1 channel (KCNQ1-C214A/G219C/C331A) before (black) and after (blue) adding 1 μM ML277. The signals after applying ML277 were normalized to the peak from the control to correct the effects of photobleaching. Scale bars are for the fluorescence signals before ML277 application. Inset shows current traces of pseudo-WT KCNQ1 channel before (black) and after (blue) adding ML277 at +40 mV. Right panels show VCF results of pseudo-WT KCNQ1 channel before (black) and after (blue) adding ML277 recorded at −60 mV, 0 mV, and 60 mV. (**B**) Fluorescence-voltage (F–V, n = 4) and G–V (n = 3) relationships of pseudo-WT KCNQ1 before (black) and after (blue) adding ML277. (**C**) A cartoon scheme to illustrate that ML277 specifically enhances E-M coupling when the VSD is at the activated state.
DOI: https://doi.org/10.7554/eLife.48576.007

results indicate that ML277 does not affect the two-step VSD activation although it doubles the current size (*Figure 6A*). The unchanged VSD activation with doubled conductance in the AO state strongly suggests that ML277 enhances the E-M coupling to increase the AO current. We have previously shown that the KCNQ1 channel may have distinct E-M coupling mechanisms to open the pore when the VSD is at the intermediate and activated state (*Hou et al., 2017*; *Zaydman et al., 2014*), thus small molecule compounds which can specifically target distinct E-M coupling pathways in KCNQ1 is not unexpected. Our results are consistent with the mechanism that ML277 may interact with a site in the KCNQ1 channel that controls E-M coupling specifically when the VSD is at the activated state.

The two open states gating processes of KCNQ1 channels can be illustrated in a scheme (*Figure 6C*) based on a simplified kinetic model without considering that the KCNQ1 channel is formed by four subunits with four identical VSDs (*Hou et al., 2017*; *Zaydman et al., 2014*). In this model, the resting closed (RC), intermediate closed (IC), and activated closed (AC) are VSD states at the resting, intermediate, and activated states when the pore is in the closed state, and IO and AO are the two open states. ML277 may specifically affect the E-M coupling of the AO state by stabilizing the conformational energy of the AO state (*Figure 6C*).

We parameterized this simple kinetic model and simulated the ML277 effects (*Figure 7*). In this model, the VSD transitions (RC↔IC↔AC) and pore transitions (IO↔AO) are voltage dependent, and the VSD-pore coupling transitions (IC↔IO and AC↔AO) are voltage independent (see Materials and methods). This model is capable of emulating the properties of the two open states in KCNQ1 channels, including the well-resolved IO and AO states and the hooked tail currents at −40 mV repolarization voltage indicating the inactivation (black, *Figure 7A*) (*Hou et al., 2017*). In this model, $k_2$ and $\beta_3$ are the transition rates for AO→AC and AO→IO respectively. To model ML277's mechanism of enhancing E-M coupling of the AO states, we reduced the rates of exiting from the AO state ($k_2$ and $\beta_3$) by half, which corresponds to ML277 stabilizing the conformational energy of the AO state (*Figure 7B*). Strikingly, reducing these rates can recapitulate the ML277 effects on KCNQ1 channel, including the enhanced AO currents and the right-shifted G-V relationship (*Figure 7A–D*, see Materials and methods).

We also prepared all-state occupancy before (black) and after (blue) adding ML277 (*Figure 7—figure supplement 1*) and found that in the model there is large reserve of unopened probability that is recruited by ML277, leading to the significant increase in the AO state occupancy and current (*Figure 7* and *Figure 7—figure supplement 1*). Such a large reserve of unopened KCNQ1 channels was also observed in single channel recordings, in which most traces were silent with no channel openings, and in those few traces that show KCNQ1 channel activity, only very short openings were observed (*Hou et al., 2017*). The large reserved occupancy of closed states provides a considerable potential to open KCNQ1 channels by activators like ML277.

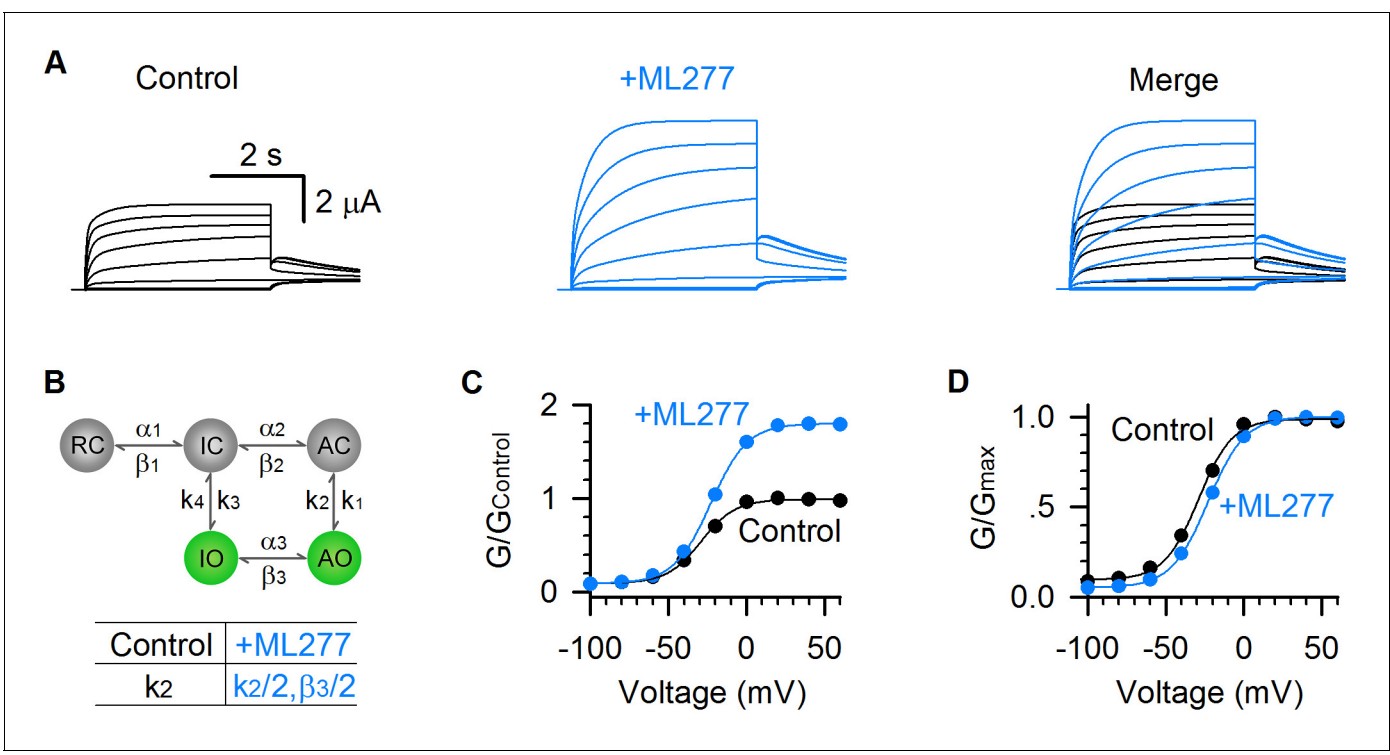

**Figure 7.** Model simulations of ML277 effects on KCNQ1 channels. (**A**) Model simulations of KCNQ1 activation currents before (black) and after (blue) adding ML277. (**B**) Five-state Markov model to show the two open states gating processes of KCNQ1 channel. α and β are voltage-dependent transitions. $k_{1-4}$ are E-M coupling rates (constant). Reducing the $k_2$ by half ($k_2$ = 853.08 for control, and $k_2$ = 426.54 for adding ML277) can mimic the ML277 effects on KCNQ1 channel. To balance the model, $\beta_3$ is also reduced by half for adding ML277 (see Materials and methods). (**C–D**) G–V relationships of simulated KCNQ1 before (black) and after (blue) adding ML277. Data points were fitted with a Boltzmann function.

DOI: https://doi.org/10.7554/eLife.48576.008

The following figure supplement is available for figure 7:

**Figure supplement 1.** Model simulations of all-state occupancy before and after adding ML277.

DOI: https://doi.org/10.7554/eLife.48576.009

Taken together, our experimental and simulation results reveal that ML277 increases KCNQ1 current by promoting the AO state E-M coupling. This is a novel gating modification mechanism that may help understand the fundamental gating processes of KCNQ1 and $I_{Ks}$ channels, and develop new strategies for treating LQTS.

## Discussion

In this work, we systematically studied the functional properties of KCNQ1 channels that are modulated by ML277, including current amplitudes and kinetics (*Figure 1*), voltage dependence of activation (*Figure 2*), inactivation (*Figure 3*), and ion permeation (*Figure 4*). All these modulations are consistent with the mechanism that ML277 specifically enhances the AO state E-M coupling to increase the AO state current.

KCNQ1 has two different open states, IO and AO. Based on recent single channel studies, there are several different levels of single channel conductance of KCNQ1 and $I_{Ks}$ channels. These levels are dependent on whether the KCNQ1 channel is associated with KCNE1 subunit and how many voltage sensors are activated (*Murray et al., 2016*; *Westhoff et al., 2019*). It is possible that when the AO state is modulated by ML277, not only the gating (VSD-pore coupling) is potentiated, but also the single channel conductance associated with the AO state is increased, and both could contribute to the increase of the AO-state-specific current.

Several lines of evidence support that ML277 changes the KCNQ1 channel gating instead of simply increasing KCNQ1 single channel conductance: (1) The current kinetics are changed upon ML277 application (*Figure 1*). (2) Previous study showed that the ML277 binding site is located away from the pore (*Xu et al., 2015*), making it hard to directly affect single channel conductance. (3) ML277 selectively enhances the current of the AO state but not that of the IO state. This is consistent with our previous finding of distinctive VSD-pore coupling mechanisms when VSD is at the intermediate and activated states, respectively (*Hou et al., 2017*), such that ML277 could modulate one mechanism without affecting the other. On the other hand, there is no known mechanism to explain how ML277 simply increases single channel conductance to selectively enhance the current amplitude of the AO state. (4) In a separate study, we found a series of residues that are critical for the VSD-pore coupling when the VSD is at the fully activated state, and ML277 interacts with these residues (data not shown). These results support that ML277 may specifically modulate VSD-pore coupling when the VSD is at the fully activated state.

Different compounds and toxins have been found as state-dependent modulators for ion channels. For instance, mexiletine preferentially blocks the $Na_V1.5$ sodium channel when the DIII-VSD is activated, and protoxin-II shows higher affinity to resting VSD states of $Na_V1.7$ sodium channel (*Nakagawa et al., 2019*; *Xu et al., 2019*; *Zhu et al., 2019*), while roscovitine facilitates the inactivation and shows less effect on activation of $Ca_V1.2$ calcium channel (*Yarotskyy et al., 2010*; *Yazawa et al., 2011*). We previously found that the IO state is more sensitive to the blocker XE991 than the AO state (*Zaydman et al., 2014*). These modulators preferentially modulate one of the states of certain ion channels but still show considerable modulations at other states. In this study, we show a clean example of state-dependent modulation in which only the AO state of KCNQ1 channel is exclusively enhanced by ML277, leaving the IO state unaltered. This unique mechanism of gating modification reveals an interesting mechanistic feature of the KCNQ1 channel. Since only the AO state is exclusively modulated, this strongly suggests that there are structural and functional modules in KCNQ1 channels that engage in interactions exclusive for the AO state E-M coupling. Moreover, these interactions can be specifically altered, when the VSD is at the activated state, to yield higher open probability. This is consistent with our previous finding that single mutations can selectively disrupt the E-M couplings for the IO and AO states, respectively (*Hou et al., 2017*; *Zaydman et al., 2014*). In a previous study, Xu and colleagues proposed that ML277 intracellularly binds at the inter-subunit interface between intracellular loops (S2-S3 loop and S4-S5 linker) and the pore (*Xu et al., 2015*). It will be interesting to further study if this putative ML277 binding site is important in the E-M coupling for the AO state.

We studied 1 µM ML277-induced functional changes to KCNQ1 channels expressed in *Xenopus* oocytes. Some previous publications reported different observations under other experimental conditions. For example, *Yu et al. (2013)* found that in CHO cells, 0.3 µM ML277 strongly increases KCNQ1 currents, left-shifts the G–V relationship, and abolishes inactivation. However, Xu et al.

reported a much less effect of ML277 on KCNQ1 activation expressed in *Xenopus* oocytes than in CHO cells (*Xu et al., 2015*). Xu et al. also found that in *Xenopus* oocytes, 25 µM ML277 strongly decreases the $Rb^+/K^+$ permeability ratio to the level even lower than $I_{Ks}$ channels (*Xu et al., 2015*). The high concentrations of ML277 used by Xu and colleagues may enhance the AO state more than KCNE1 does to the channel, or the specificity of ML277 at this high concentration might have changed.

Activators of KCNQ1 and $I_{Ks}$ channels may be effective for treating LQTS. Although several KCNQ1 activators have been identified, their low specificity may limit further drug development (*Cruickshank et al., 2003*; *De Silva et al., 2018*; *Gao et al., 2017*; *Hill and Sitsapesan, 2002*; *Mruk and Kobertz, 2009*; *Salata et al., 1998*; *Zheng et al., 2012*). Several features of ML277 make it an interesting KCNQ1 modulator. First, ML277 specifically activates KCNQ1 channel with little activity on other ion channels that are key to cardiac electrophysiology or neuronal KCNQ channels (*Mattmann et al., 2012*; *Yu et al., 2013*). Second, ML277 activates KCNQ1 channels expressed in mammalian cells with an $EC_{50}$ of 260 nM (*Mattmann et al., 2012*; *Yu et al., 2013*), lower than any other known KCNQ1/$I_{Ks}$ activators. Finally, studies have confirmed that ML277 effectively shortens the APD in human iPSCderived cardiomyocytes (including LQTS conditions) and guinea pig cardiomyocytes (*Ma et al., 2015*; *Xu et al., 2015*; *Yu et al., 2013*). Interestingly, ML277 was found to increase only the current of KCNQ1 channels but not the $I_{Ks}$ channels with saturated KCNE1 association in exogenous expression (*Figure 1A,B*) (*Yu et al., 2013*). The sensitivity of $I_{Ks}$ currents to ML277 in cardiomyocytes suggests that native $I_{Ks}$ channels are not always saturated with KCNE1 association, and these native $I_{Ks}$ channels may be a novel target for compounds such as ML277 to treat LQTS.

## Materials and methods

### Constructs and mutagenesis

Overlap extension and high-fidelity PCR were used for making KCNQ1 channel point mutations. Each KCNQ1 mutation was verified by DNA sequencing. Then cRNA of WT KCNQ1 and mutants were synthesized using the mMessage T7 polymerase kit (Applied Biosystems-Thermo Fisher Scientific) for oocyte injections.

### Oocyte expression

Oocytes (at stage V or VI) were obtained from *Xenopus laevis* by laparotomy surgery, following the protocol approved by the Washington University Animal Studies Committee (Protocol #20160046). After six oocyte removal operations, the frog was anesthetized in 1.5L stagnant tap water containing 2 g Ethyl 3-aminobenzoate, methanesulfonic acid salt (Acros Oranics 886-86-2), buffered with 1.5 g of NaHCO3 (Sigma S5761) for about 30 min. The heart was cut off when the frog was under fully anesthesia. All procedures are consistent with the recommendations of the Panel on Euthanasia of the American Veterinary Medical Association.

Oocytes were then digested by collagenase (0.5 mg/ml, Sigma Aldrich, St Louis, MO) and micro-injected with KCNQ1 cRNAs. WT or mutant KCNQ1 cRNAs (9.2 ng) with or without KCNE1 cRNA (0.031 ng for KCNQ1:KCNE1 weight ratio of 300:1 or 2.3 ng for KCNQ1:KCNE1 weight ratio of 4:1) were injected into each oocyte. Injected cells were kept in ND96 solution (in mM): 96 NaCl, 2 KCl, 1.8 $CaCl_2$, 1 $MgCl_2$, 5 HEPES, 2.5 $CH_3COCO_2Na$, 1:100 Pen-Strep, pH 7.6) at 18°C for 2–6 days for electrophysiology recordings.

### Two-electrode voltage clamp (TEVC) and voltage-clamp fluorometry (VCF)

Microelectrodes (Sutter Instrument, Item #: B150-117-10) were made with a Sutter (P-97) puller with resistances between 0.5 MΩ and 3 MΩ when filled with 3 M KCl. After channel expression, oocyte cells were transferred to the recording chamber in ND96 bath solutions for whole-oocyte currents recording. Currents, sampled at 1 kHz and low-pass-filtered at 2 kHz, were recorded using a CA-1B amplifier (Dagan, Minneapolis, MN) with Patchmaster (HEKA) acquisition software. All recordings were performed at room temperature (21–23°C). For ML277 experiments, ML277 stock (Sigma Aldrich, 1 mM in DMSO) was added to the bath and diluted to 1 µM. For the ion permeation

experiments, 100 mM $K^+$ and 100 mM $Rb^+$ solutions (96 mM NaCl was replaced with 100 mM KCl and 100 mM RbCl from ND96 solution) were perfused onto cells to steady state.

For VCF experiments, oocytes were incubated in 10 μM Alexa 488 C5-maleimide (Molecular Probes, Eugene, OR) on ice for 30 min. To facilitate the labeling, Alexa 488 was prepared into high $K^+$ solution in mM (98 KCl, 1.8 $CaCl_2$, 5 HEPES, pH 7.6) to depolarize the membrane voltage so that the VSD could undergo outward movement. After 30 min, oocytes were transferred to normal ND96 solution and washed three times before recording. The excitation and emission lights were filtered for Alexa 488. During recording, the fluorescence signals from the VSD movements were collected by a photodiode (Pin20A, OSI Optoelectronics). The signals, sampled at 1 kHz and filtered at 200 Hz, were then amplified by an EPC10 (HEKA) amplifier and synchronized by the CA-1B amplifier.

## Electrophysiology data analysis

Data were analyzed with Clampfit (Axon Instruments, Inc, Sunnyvale, CA), Sigmaplot (SPSS, Inc, San Jose, CA), and IGOR (Wavemetrics, Lake Oswego, OR). Because of photo-bleaching, fluorescence signals were baseline subtracted by fitting the first 2 s signals at the −80 mV holding potential. Fluorescence-voltage (F-V) relationships were derived by normalizing the $\Delta F/F$ value at the end of each test pulse to the maximal value. G–V and F–V curves were fitted with either one or double Boltzmann equations in the form of $1/(1 + \exp(-z*F*(V-V_{1/2})/RT))$, where $V$ is the voltage, $z$ is the equivalent valence, $V_{1/2}$ is the half-maximal voltage, $F$ is the Faraday constant, $R$ is the gas constant, and $T$ is the absolute temperature. KCNQ1 tail currents were fitted with a double exponential function $F(t) = A_1 \times \exp(-t/\tau_1) + A_2 \times \exp(-t/\tau_2) + C$ as previously described (*Hou et al., 2017*) to quantify the hook currents ($A_2$). For activation time constants ($\tau_f$ and $\tau_s$) of KCNQ1 currents in *Figure 2*, KCNQ1 activation currents were fitted with a double exponential function to get the $\tau_f$ and $\tau_s$ for control, and were fitted separately with a single exponential function to get the $\tau_f$ and $\tau_s$ for currents after adding ML277.

## Statistics

All averaged data were collected from at least three different cells. Pairwise comparison in *Figure 2C*, between Control and +ML277 groups, was achieved using paired t test (n = 25) with a significant difference ($p=5\times10^{-10}$). Other comparison between any two data groups was achieved using t test. Raw data points were provided on top of the bar presentations. All error bars represent standard error of the mean.

## Kinetic modeling of ML277 effects

The five-state kinetic model of $K_V7.1$ channel was constructed from our previous six-state model (*Hou et al., 2017*). The upper row RC, IC, AC stand for the VSD states at resting, intermediate, and activated when the pore is closed, and the lower row IO and AO stand for pore opening at intermediate and activated states. The voltage independent opening state 'resting open' (RO) was deleted from the six-state model based on the rare transition from RC to RO (*Hou et al., 2017*). This simplified five-state model is capable to recapitulate the main characteristics of the $K_V7.1$ two open states gating (*Hou et al., 2017*). We use the same Markov process to model the transition between each state, where $\alpha_1 = a_1*\exp(v/m)$, $\beta_1 = c_1*\exp(-v/n)$, $\alpha_i = a_i*\exp(v/b)$ and $\beta_i = c_i*\exp(-v/d)$ (i = 2 and 3) are the voltage-dependent rates of transitions. $k_{1-4}$ are the E-M coupling rates (constant), where $k_2$, that is the E-M coupling rate from the AO state to the AC state, can be set to different values to simulate the control and after ML277 results. Reducing the value of $k_2$ by half can mimic the ML277 effects on KCNQ1 channel. The values of the parameters are as follows: $a_1 = 0.00070$ $ms^{-1}$, $a_2 = 0.0047$ $ms^{-1}$, $a_3 = 0.15$ $ms^{-1}$, $c_1 = 0.0020$ $ms^{-1}$, $c_2 = 0.00017$ $ms^{-1}$, $c_3 = 0.048$ $ms^{-1}$, m = 46.0 mV, n = 31.2 mV, b = 37.7 mV, d = 41.5 mV, $k_1 = 0.89$, $k_2 = 853.08$ (for control), and $k_2 = 426.54$ (for adding ML277), $k_3 = 0.96$, $k_4 = 103.82$. A kinetic model with a closed loop needs to obey the detailed balance (*Zhang et al., 2016*). To balance the model, $c_3 = 0.024$ $ms^{-1}$ for adding ML277. Kinetic parameters were optimized with CeL software (HUST, Wuhan, Hubei, China) as previously described (*Hou et al., 2014*; *Wang et al., 2013*). The reversal potential was set to −80 mV, and the single-channel conductance was set to 0.18 pS for KCNQ1 channels (*Hou et al., 2017*). This is a simple model which may not be able to reproduce all the aspects modified by ML277, for example the slowing kinetics of the activation (*Figure 1*) and the deactivation (*Figure 3*). It is likely that besides

enhancing the AO state VSD-pore coupling, ML277 has an additional effect to specifically modulate the AO state gating kinetics, which the model cannot reproduce this additional effect.

## Acknowledgements

We thank Po Wei Kang for reading and revising the manuscript. We also thank Prof. Jeanne Nerbonne for helpful discussion. This work was supported by R01 NS092570 and R01 HL126774 (to JC), and by AHA 18POST34030203 (to PH).

## Additional information

### Competing interests

Jingyi Shi, Jianmin Cui: is a co-founder of a startup company VivoCor LLC, which is targeting IKs for the treatment of cardiac arrhythmia. The other authors declare that no competing interests exist.

### Funding

| Funder | Grant reference number | Author |
| --- | --- | --- |
| National Institute of Neurological Disorders and Stroke | R01 NS092570 | Jianmin Cui |
| National Heart, Lung, and Blood Institute | R01 HL126774 | Jianmin Cui |
| American Heart Association | AHA 18POST34030203 | Panpan Hou |

The funders had no role in study design, data collection and interpretation, or the decision to submit the work for publication.

### Author contributions

Panpan Hou, Conceptualization, Data curation, Formal analysis, Supervision, Funding acquisition, Investigation, Methodology, Writing—original draft, Writing—review and editing; Jingyi Shi, Kelli McFarland White, Data curation, Formal analysis; Yuan Gao, Data curation, Formal analysis, Investigation; Jianmin Cui, Supervision, Funding acquisition, Investigation, Writing—original draft, Writing—review and editing

### Author ORCIDs

Panpan Hou  https://orcid.org/0000-0001-7694-2262
Yuan Gao  https://orcid.org/0000-0002-7661-2367
Jianmin Cui  https://orcid.org/0000-0002-9198-6332

### Ethics

Animal experimentation: Oocytes (at stage V or VI) were obtained from *Xenopus laevis* by laparotomy surgery, following the protocol approved by the Washington University Animal Studies Committee (Protocol #20160046).

### Decision letter and Author response

Decision letter https://doi.org/10.7554/eLife.48576.012
Author response https://doi.org/10.7554/eLife.48576.013

## Additional files

### Supplementary files

• Transparent reporting form
DOI: https://doi.org/10.7554/eLife.48576.010

## Data availability

All data generated or analysed during this study are included in the manuscript.

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
