## [Decision Letter]

Thank you for submitting your article "ML277 specifically enhances the fully activated open state of KCNQ1 by modulating VSD-pore coupling" for consideration by *eLife*. Your article has been reviewed by three peer reviewers, one of whom is a member of our Board of Reviewing Editors, and the evaluation has been overseen by Olga Boudker as the Senior Editor. The reviewers have opted to remain anonymous.

The reviewers have discussed the reviews with one another and the Reviewing Editor has drafted this decision to help you prepare a revised submission. Please aim to submit the revised version within two months.

Summary:

KCNQ and its accessory subunit KCNE contribute to the IKs currents in the heart. Mutations of these channels cause LQT syndrome and this channel is a target for the development of anti-arrhythmic drugs. Previously, Cui and colleagues have shown that the KCNQ channel contributes to two distinct open states depending on the presence or absence of the KCNE subunit. In presence of KCNE the AO state is stabilized whereas the channel prefers to open to IO state in absence of this accessory subunit. In this study, Hou et al. use a combination of electrophysiology, kinetic analysis, mutagenesis, and voltage clamp fluorometry to show that ML277 enhances current by specifically modulating the AO state of KCNQ channels. Kinetic analysis of KCNQ activation in the presence of ML277 shows that this compound selectively enhances the slow component of activation. This effect is obscured when KCNQ is co-expressed with saturating levels of the accessory subunit KCNE1 suggesting that the drug has the same effect on the channel as KCNE1. The authors go on to show that ML277 influences the inactivation and permeability of KCNQ in a manner consistent with selective action on the AO state. Next, they use several mutants which specifically stabilize the IO (E1R/R2E and S338F) or AO (E1R/R4E and F351A) states to more directly probe the influence of this compound on the distinct open states. Finally, VCF and kinetic modeling are provided as support to their model in which ML277 binding enhances conductance by binding to the AO state and enhancing coupling. Overall, this study highlights a new mechanism of drug action whereby the drug selectively stabiles the channel in one open state and acts as a surrogate for the KCNE subunit. Nevertheless, the reviewers had a number of concerns with regards to the interpretation of the experimental data.

Essential revisions:

1) The authors provide convincing evidence that the modulation is specific to the AO state (using mutagenesis and modeling) and that the drug does not influence voltage sensing (through VCF). The authors interpret these two points to indicate that ML277 increases the open probability of the AO state via enhanced coupling between VSD and pore. However, an alternative explanation for this data is that ML277 increases the unitary conductance of the AO state, and no experiments are provided to rule this out. The overall effect of ML277 on KCNQ channels both resembles the effect of KCNE subunit and is abrogated by KCNE. The authors have previously shown that KCNE increases the unitary conductance 10 fold (Hou et al., 2017). Distinguishing between an influence on open probability and single channel conductance may be beyond the scope of this work but the manuscript will certainly benefit from a more balanced discussion.

2) I wonder why ML277 is not effective on KCNQ1/KCNE1 channel current (Figure 1B), if it changes the AO state E-M coupling. In F351A mutant (Figure 5B) which opens only in the AO state, similarly to the case of KCNQ1/KCNE1 channel, a clear effect of ML277 is observed.

3) The study would benefit from a clearer description of the model outcomes (what is the state occupancy before and after ML277 in the model). The model parameters suggest there is large reserve of unopened channels that are then recruited by ML277, leading to the significant current increase in ML277. Is this a justifiable feature of the model?

Subsection “Kinetic modeling of ML277 effects” and Figure 7 legend: It is written that the model c_3_ (β_3_) was reduced to half to balance the model. As the meaning here is not clear, please explain in more detail. Why did the authors try only [both 1/2 of k_2_ and 1/2 of c_3_]? Are there any other possibilities? e.g. What will happen if only k_2_ is reduced to half?

4) Based on the state diagram in Figure 7B, I do not fully understand why the deactivation kinetics in Figure 3C is slowed down by ML277. I understand that channels in AO state enter IO state after a short time upon hyperpolarization, and then go back to IC state. If this is the case, the deactivation kinetics would be expected to be unchanged. Was the slowing of the deactivation kinetics by MLL277 reproduced by model simulation?

5) Figure 2A, C: Is there any change of the kinetics of recovery from inactivation (i.e. AO- IO transition, i.e. kinetics of the increasing phase in the hook current) by ML277?

6) Data surrounding the description of Figure 1B should be more clearly rationalized. The saturated Q1:E1 complex is described to exclusively activate to the A0 state, which I understood the paper to argue might be stabilized by ML277. There is no indication that this takes place in the 4:1 IKs complex (for example, no slowing of tail currents, as predicted in the model if I understand correctly).

7) Statistical analysis was done only for Figure 2C. I suggest to perform it also in other data such as Figures 1C, D, 3D, 6B, 4B, 5C, 7D etc., and to describe the detail of the statistical analysis for them in the Materials and methods section. I suggest to indicate S.D. values, not S.E, and to present all of each data point, as well as mean and S.D values, rather than to use bar presentations.

---

## [Author Response]

Essential revisions:1) The authors provide convincing evidence that the modulation is specific to the AO state (using mutagenesis and modeling) and that the drug does not influence voltage sensing (through VCF). The authors interpret these two points to indicate that ML277 increases the open probability of the AO state via enhanced coupling between VSD and pore. However, an alternative explanation for this data is that ML277 increases the unitary conductance of the AO state, and no experiments are provided to rule this out. The overall effect of ML277 on KCNQ channels both resembles the effect of KCNE subunit and is abrogated by KCNE. The authors have previously shown that KCNE increases the unitary conductance 10 fold (Hou et al., 2017). Distinguishing between an influence on open probability and single channel conductance may be beyond the scope of this work but the manuscript will certainly benefit from a more balanced discussion.

We thank the reviewers for raising this key point, and appreciate the constructive suggestions. We follow the suggestion of the reviewers to make a more thorough discussion on this issue in the revised manuscript. The discussion is focused on the following aspects.

KCNQ1 has two open states, IO and AO. Based on Dr. David Fedida’s recent single channel results, there are several different levels of single channel conductance of KCNQ1 and IKs channels. These levels are dependent on whether the KCNQ1 channel is associated with KCNE1 subunit, and how many voltage sensors are activated (Murray et al., 2016; Westhoff et al., 2019). It is possible that when the AO state is modulated by ML277, not only the gating (VSD-pore coupling) is potentiated, but also the single channel conductance associated with the AO state is increased, and both could contribute to the increase of the AO state specific current.

We appreciate the recognition by the reviewers that distinguishing between an influence on open probability and single channel conductance may be beyond the scope of this work. To substantiate this concern, we would like to point out that, first of all, it is technically difficult to record KCNQ1 single channel currents. Even for Dr. David Fedida’s lab, which is the best on this kind of studies, it may take a long time to get some data from our experience in collaborating with him (Hou et al., 2017). On the other hand, single channel current amplitude may increase once AO is potentiated, as discussed above. Therefore, single channel recordings may not resolve the contribution of gating and conductance to the increased currents by ML277.

We have also thought about using noise analysis to get the KCNQ1 single channel conductance before and after ML277 (Yang and Sigworth, JGP 1998; Sesti and Goldstein, JGP 1998). However, this method assumes that the channel has open-close two states. Since KCNQ1 channels not only have two open states but also have different sub-conductance levels, this approach is not a feasible solution. In addition, previous studies show that the KCNQ1 channel single channel openings are very short, which will also compromise the noise analysis.

On the other hand, we think that the following evidences support that ML277 changes the KCNQ1 channel gating: 1) The current kinetics are changed upon ML277 application (Figure 1); 2) Previous study showed that the ML277 binding site is located away from the pore (Xu et al., 2015), making it hard to directly affect single channel conductance; 3) ML277 selectively enhances the current of the AO state but not that of the IO state. This is consistent with the previous finding of distinctive VSD-pore coupling mechanisms when VSD is at the intermediate and activated states, respectively (Hou et al., 2017), such that ML277 could modulate one mechanism without affecting the other. On the other hand, there is no known mechanism to explain how ML277 simply increases single channel conductance to selectively enhance the current amplitude of the AO state; 4) In a separate study, we found a series of residues that are critical for the VSD-pore coupling when the VSD is at the fully activated state, and ML277 interacts with these residues. These results support that ML277 may specifically modulate VSD-pore coupling when the VSD is at the fully activated state.

2) I wonder why ML277 is not effective on KCNQ1/KCNE1 channel current (Figure 1B), if it changes the AO state E-M coupling. In F351A mutant (Figure 5B) which opens only in the AO state, similarly to the case of KCNQ1/KCNE1 channel, a clear effect of ML277 is observed.

Based on other experimental and simulation studies (Xu et al., 2015; Yu et al., 2013) and our unpublished data, ML277 may bind at the interface between the VSD and pore formed by two neighboring subunits, where KCNE1 is also found to bind with KCNQ1 (Chan et al., 2012; Nakajo et al., 2010; Peng et al., 2017; Ramasubramanian and Rudy, 2018; Strutz-Seebohm et al., 2011; Xu and Rudy, 2018). As a result, KCNE1 binding may block the ML277 binding site. Therefore, ML277 shows no effect on KCNQ1/KCNE1 channels with saturated KCNE1 binding.

In the case of F351A mutant, it opens only in the AO state but leaves the ML277 binding site intact. Therefore, a clear current increase is observed when adding ML277. We have added some sentences to clarify this point more clearly in the revised manuscript.

3) The study would benefit from a clearer description of the model outcomes (what is the state occupancy before and after ML277 in the model). The model parameters suggest there is large reserve of unopened channels that are then recruited by ML277, leading to the significant current increase in ML277. Is this a justifiable feature of the model?

The reviewers are correct. In the model there is a large reserve of unopened probability that is recruited by ML277, leading to the significant current increase. We prepared all-state occupancy before (black) and after (blue) adding ML277, and it shows clearly that by reducing the rates of exiting from the AO state (k_2_ and β_3_) by half, ML277 specifically increases the AO state occupancy (Figure 7—figure supplement 1). From the model, the reserve is determined by the VSD-pore coupling, i.e., the vertical transitions (k factors). With these parameters (k_1_= 0.89, k_2_= 853.08, k_3_= 0.96, k_4_= 103.82), the majority of the channels are at the AC and IC states even at high voltages, leaving a considerable room for IO and AO states to be modified.

Such a large reserve of unopened KCNQ1 channels was observed in single channel recordings, in which most traces were silent with no channel openings, and in those few traces that show KCNQ1 channel activity, only very short openings were observed (Hou et al., 2017). Our previous studies suggest that the reserved occupancy can be altered. We found that PIP2 is important for the VSD-pore coupling (Zaydman et al., 2013), and the IO state has less PIP2 sensitivity than the AO state (Li et al. PNAS 2013; Zaydman et al., 2013). In the case of KCNQ1 alone, which primarily opens at the IO state with low PIP2 sensitivity, native PIP2 in oocytes is not sufficient to fully activate the channel. The total current increases dramatically when high concentration PIP2 is applied from the intracellular side of the membrane (Li et al. PNAS 2013). This is a good point raised by the reviewers. We have added the occupancy of the states with and without ML277 as a new Figure 7—figure supplement 1, and discussed this point in the revised manuscript.

Subsection “Kinetic modeling of ML277 effects” and Figure 7 legend: It is written that the model c_3_ (β_3_) was reduced to half to balance the model. As the meaning here is not clear, please explain in more detail. Why did the authors try only [both 1/2 of k_2_ and 1/2 of c_3_]? Are there any other possibilities? e.g. What will happen if only k_2_ is reduced to half?

It is required for detailed balance that to reduce k_2_, another rate constant has to be changed as well. This other rate constant does not have to be c_3_ (β_3_). For instance, ½ k_2_ and ½ k_4_ also would maintain detailed balance. However, we chose to reduce β_3_ because this rate is for exiting the AO state (to reserve more of the AO state). We did not do a good job in the original manuscript to explain this. We have explained it in more detail in the revised manuscript.

For the requirement of detailed balance, we quote Zhang et al., 2016: “A hidden Markov model with detailed balance is crucial to modeling transitions at equilibrium. Detailed balance is both a necessary and a sufficient condition for any system in thermal equilibrium. Detailed balance requires the probability flux from state *i* to state *j* to be equal to the flux in a reverse direction, or *p_i_q_ij_ = p_j_q_ji_*, for any pair of states, where *p_i_* is the probability being in state *i*, and *q_ij_* is the rate constant of the transition from state *i* to state *j*. As a result, any net probability flux in the reaction network, including steady-state flux along any closed loop, is prohibited.”

In our model, the states IC, AC, AO and IO form a closed loop (Author response image 1), and factors k_1_ and k_2_ are primarily controlling the AO coupling. To mimic the ML277 effects, enhancing the AO state coupling, we reduced the AO→AC rate k_2_ by half (reduce exiting rate to reserve more of the AO state). To balance the model, we also reduced the AO→IO rate β_3_ by half. These changes can recapitulate the ML277 effects on KCNQ1 channel, including the enhanced AO currents and the right-shifted G-V relationship (Figure 7A-D).

Alternatively, we can do 2X of k_1_ (to facilitate entering the AO state) and 1/2 of β_3_ (to balance the model) to mimic the ML277 effects (Author response image 1). The results are very similar as shown in Figure 7.

**Author response image 1. respfig1:** Model simulations of ML277 effects by 2X k_1_ and 1/2X β_3_. (**A**) Model simulations of KCNQ1 activation currents before (black) and after (blue) adding ML277. (**B**) Five-state Markov model to show the two open states gating processes of KCNQ1 channel. α and β are voltage-dependent transitions. k_1-4_ are VSD-pore coupling rates (constant). 2X K_1_ and 1/2X β_3_ can mimic the ML277 effects on KCNQ1 channel. (**C-D**) G–V relationships of simulated KCNQ1 before (black) and after (blue) adding ML277. Data points were fitted with a Boltzmann function.

4) Based on the state diagram in Figure 7B, I do not fully understand why the deactivation kinetics in Figure 3C is slowed down by ML277. I understand that channels in AO state enter IO state after a short time upon hyperpolarization, and then go back to IC state. If this is the case, the deactivation kinetics would be expected to be unchanged. Was the slowing of the deactivation kinetics by MLL277 reproduced by model simulation?

The reviewers are right, the model doesn’t show the slowing of the deactivation kinetics by ML277 (Author response image 2). We still don’t understand why the experimental data show the slowing activation (Figure 1) and slowing deactivation (Figure 3). It is likely that besides enhancing the AO state VSD-pore coupling, ML277 has an additional effect to specifically modulate the AO state gating kinetics, but the model cannot reproduce this additional effect. This is a simple model which may not be able to reproduce all the aspects modified by ML277. We have pointed this discrepancy out in the revised manuscript.

**Author response image 2. respfig2:** Model simulations of deactivation kinetics by ML277. Model simulations of KCNQ1 channel before (black) and after (blue) adding ML277. The activation voltage was +40 mV and then stepped to −120 mV to induce a hook at the tail current. The insets show the tail currents with an expanded time scale. The time course of the deactivation currents was shown as τ. No obvious change of the deactivation kinetics was observed by ML277.

5) Figure 2A, C: Is there any change of the kinetics of recovery from inactivation (i.e. AO- IO transition, i.e. kinetics of the increasing phase in the hook current) by ML277?

We analyzed the increasing phase in the hook current (τ_2_, Author response image 3), and found that there was no significant change of the kinetics of the recovery from inactivation by ML277 (Author response image 3).

**Author response image 3. respfig3:** Kinetics of the hook current before and after adding ML277. (**A**) KCNQ1 tail current recorded with a 2.913 s pre-pulse duration (red) was fitted with a double exponential function *F(t*) = *A*_1_×exp(−*t/τ*_1_)+ *A*_2_×exp(−*t/τ*_2_)+ *C* (Hou et al., 2017). The fitting curve is shown in dotted green line, the deactivation component (*A*_1_×exp(−*t/τ*_1_)+*C*) and hook component (*A*_2_×exp(−*t/τ*_2_)) are shown in green lines. (**B**) Tail currents in Figure 3A, C were fitted with a double exponential function and time constants of the hook current before (black) and after (blue) adding ML277 were plotted over pre-pulse durations.

6) Data surrounding the description of Figure 1B should be more clearly rationalized. The saturated Q1:E1 complex is described to exclusively activate to the A0 state, which I understood the paper to argue might be stabilized by ML277. There is no indication that this takes place in the 4:1 IKs complex (for example, no slowing of tail currents, as predicted in the model if I understand correctly).

We apologize for the confusing labeling in Figure 1B. The Q1:E1 = 300:1 or 4:1 indicates KCNQ1+KCNE1 channels with injected mRNA weight ratio of KCNQ1:KCNE1=300:1 or 4:1. In the 300:1 case, the IKs complex associates with unsaturated KCNE1, while in the 4:1 case, the IKs channel associates with saturated KCNE1 (Nakajo et al., 2010).

As mentioned in the answer to question 2, KCNE1 binding may block the ML277 binding site, and therefore ML277 shows no effect on KCNQ1/KCNE1 channel with saturated KCNE1 binding (Xu et al., 2015; Yu et al., 2013), which was labeled as Q1:E1 = 4:1 in our previous Figure 1B. We have changed the labeling to “unsaturated” and “saturated” in the new Figure 1B, and described more details in the figure legend.

7) Statistical analysis was done only for Figure 2C. I suggest to perform it also in other data such as Figures 1C, D, 3D, 6B, 4B, 5C, 7D etc., and to describe the detail of the statistical analysis for them in the Materials and methods section. I suggest to indicate S.D. values, not S.E, and to present all of each data point, as well as mean and S.D values, rather than to use bar presentations.

As the reviewers suggested, we have performed t-test in other figures, and described the detail in the Materials and methods. We have also provided all of each data point on top of the bar presentations.

We respectfully insist that we prefer S.E over S.D. S.D. reflects the variability of the original data points; while S.E. is the variability of the mean values. S.E. is usually used in constructing confidence intervals (CI), which indicate a range of values within which the “true” value lies. Therefore, CI shows the readers how accurate the estimates of the population values actually are (David L Streiner, Can J Psychiatry 1996;41:498–502). Based on this, we prefer to use S.E. for statistical analysis.